# Case Series of Primaquine-Induced Haemolytic Events in Controlled Trials with G6PD Screening

**DOI:** 10.3390/pathogens12091176

**Published:** 2023-09-19

**Authors:** Ayleen Kosasih, Robert James, Nguyen Hoang Chau, Michelle M. Karman, Lydia Visita Panggalo, Lyndes Wini, Ngo Viet Thanh, Thomas Obadia, Ari Winasti Satyagraha, Puji Budi Setia Asih, Din Syafruddin, Walter R. J. Taylor, Ivo Mueller, Inge Sutanto, Harin Karunajeewa, Ayodhia Pitaloka Pasaribu, J. Kevin Baird

**Affiliations:** 1Oxford University Clinical Research Unit Indonesia, Jakarta 10430, Indonesia; akosasih@oucru.org (A.K.); mkarman@oucru.org (M.M.K.); kbaird@oucru.org (J.K.B.); 2Walter and Eliza Hall Institute of Medical Research, Melbourne, VIC 3052, Australia; robert.james@miwatj.com.au (R.J.); mueller@wehi.edu.au (I.M.); 3Department of Medical Biology, University of Melbourne, Melbourne, VIC 3010, Australia; 4Oxford University Clinical Research Unit, Hospital for Tropical Diseases, District 5, Ho Chi Minh City 749000, Vietnam; chaunh@oucru.org (N.H.C.); thanhnv@oucru.org (N.V.T.); 5Exeins Health Initiative, Jakarta 12870, Indonesia; lydiavisita@gmail.com (L.V.P.); ariwsatyagraha@gmail.com (A.W.S.); 6Vector-Borne Disease Control (VBDC) Division, Solomon Islands Ministry of Health and Medical Services, Honiara P.O. Box R113, Solomon Islands; lyndes.wini@gmail.com; 7Institut Pasteur, Université Paris Cité, Bioinformatics and Biostatistics Hub, F-75015 Paris, France; thomas.obadia@pasteur.fr; 8Institut Pasteur, Université Paris Cité, G5 Infectious Diseases Epidemiology and Analytics, F-75015 Paris, France; 9Eijkman Research Center for Molecular Biology, National Research and Innovation Agency, Cibinong 16911, Indonesia; puji_bsa@yahoo.com (P.B.S.A.); dinkarim@yahoo.com (D.S.); 10Department of Parasitology, Faculty of Medicine, Hasanuddin University, Makassar 90245, Indonesia; 11Hasanuddin University Medical Research Center, Makassar 90245, Indonesia; 12Mahidol-Oxford Tropical Medicine Research Unit, Faculty of Tropical Medicine, Mahidol University, Bangkok 10400, Thailand; bob@tropmedres.ac; 13Centre for Tropical Medicine and Global Health, Nuffield Department of Medicine, University of Oxford, Oxford OX3 7BN, UK; 14Department of Parasitology, Faculty of Medicine, University of Indonesia, Jakarta 10430, Indonesia; sutanto.inge@yahoo.com; 15Department of Medicine, Western Health, The University of Melbourne, Melbourne, VIC 3010, Australia; harin.karunajeewa@wh.org.au; 16Department of Pediatrics, Medical Faculty, Universitas Sumatera Utara, Medan 20155, Indonesia

**Keywords:** acute haemolytic anaemia, primaquine, G6PD deficiency, G6PD screening, randomized controlled trials

## Abstract

Primaquine for radical cure of *Plasmodium vivax* malaria poses a potentially life-threatening risk of haemolysis in G6PD-deficient patients. Herein, we review five events of acute haemolytic anaemia following the administration of primaquine in four malaria trials from Indonesia, the Solomon Islands, and Vietnam. Five males aged 9 to 48 years were improperly classified as G6PD-normal by various screening procedures and included as subjects in trials of anti-relapse therapy with daily primaquine. Routine safety monitoring by physical examination, urine inspection, and blood haemoglobin (Hb) assessment were performed in all those trials. Early signs of acute haemolysis, i.e., dark urine and haemoglobin drop >20%, occurred only after day 3 and as late as day 8 of primaquine dosing. All patients were hospitalized and fully recovered, all but one following blood transfusion rescue. Hb nadir was 4.7 to 7.9 g/dL. Hospitalization was for 1 to 7 days. Hb levels returned to baseline values 3 to 10 days after transfusion. Failed G6PD screening procedures in these trials led G6PD-deficient patients to suffer harmful exposures to primaquine. The safe application of primaquine anti-relapse therapy requires G6PD screening and anticipation of its failure with a means of prompt detection and rescue from the typically abrupt haemolytic crisis.

## 1. Introduction

*Plasmodium vivax* is widespread in many regions of the world and is particularly prevalent in countries in South Asia, Southeast Asia, Oceania, and parts of South America [1]. It is estimated that this species caused 4.9 million cases in 2021 globally [1]. Its ability to form a dormant stage in the liver poses a great challenge to detect and target this species for intervention [2]. Infections arising from re-awakened hypnozoites are deemed to be the most problematic for the elimination program [2]. It has been estimated that 66–96% of all *P. vivax* blood-stage infections are due to relapses [3].

Latency in *Plasmodium vivax* malaria may be cured only with the administration of 8-aminoquinoline chemotherapeutics, namely primaquine or tafenoquine [4,5]. This class of drugs invariably causes dose-dependent acute haemolytic anaemia in patients having glucose-6-phosphate dehydrogenase (G6PD) deficiency [6]; this is the most common inborn human abnormality, affecting an average of 8% of residents of malaria-endemic countries, with prevalence varying from 1–35% [7]. In Indonesia, the prevalence was reported to be 2.6% in North Sumatra [8] and 3.8% in Sumba [9]. In Vietnam, the prevalence was 8.9% [10]. In the Solomon Islands, the prevalence was 20.3% [11]. Healthcare providers managing *P. vivax* patients of unknown G6PD status must choose between a therapy inviting the risk of acute haemolytic anaemia or withholding that therapy and inviting subsequent relapses and onward transmission. This clinical dilemma is mitigated by ascertaining the patient’s G6PD status, thus protecting the G6PD-deficient minority from toxic exposure to 8-aminoquinoline therapy and protecting the G6PD-normal majority from ongoing latency and often multiple attacks of malaria.

Screening for G6PD deficiency in most malaria-endemic settings has proven impractical until very recently [12]. The gold-standard quantitative spectrophotometric assay requires a laboratory equipped with sophisticated instruments operated by highly skilled technicians. A standard qualitative fluorescent spot test involves cold transport and storage of labile reagents and a laboratory setting. These requirements are prohibitive in healthcare facilities in remote rural locations where most of the burden of *P. vivax* malaria occurs. Point-of-care G6PD testing technologies aimed at solving this serious problem of access have recently emerged [12,13]. The availability of these in routine care will provide safer access to 8-aminoquinoline therapies against repeated relapses [14]. The WHO strongly recommends the examination of G6PD status before administering primaquine as a radical cure for *Plasmodium vivax* [4].

In this report, we highlight the risk of primaquine-induced acute haemolytic anaemia despite access to G6PD screening, even in the rigorous setting of controlled clinical trials. This retrospective case series reviews five severe adverse haemolytic events that occurred in four different randomized controlled trials (RCT) conducted in Indonesia (two subjects in two trials), the Solomon Islands (two subjects, one trial), and Vietnam (one subject, one trial) between 2012 and 2022. The authors of this report are the investigators of those trials. In all those trials, subjects with confirmed or suspected patent *P. vivax* malaria were administered varied regimens of primaquine in combination with varied blood schizontocidal therapies as a radical cure of latent malaria in order to prevent relapses. All subjects were screened for G6PD deficiency prior to enrolment using varied screening kits and technologies. All subjects were misclassified as G6PD-normal and experienced onset of severe acute haemolytic anaemia 4 to 8 days after initiation of primaquine therapy. We describe here the clinical course of these haemolytic crises, treatment, and recovery.

## 2. Case Series

### 2.1. Case 1

The subject was a 9-year-old boy enrolled in a randomized controlled trial involving serological screening for exposure to *P. vivax* malaria and primaquine therapy against latency in North Sumatra, Indonesia (reg no. NCT04223674). A total of 1133 participants aged 6–15 years old were enrolled from February 2022–May 2023. Serological screening was performed to predict recent *P. vivax* exposure, which may indicate the risk of harbouring hypnozoites for which primaquine was given as a radical cure to prevent future relapse. These participants were recruited in a school-based mass blood survey. Physical examination by the study physician and capillary blood collection from finger prick was done onsite, whereas Hb and G6PD screening was performed at the nearby site laboratory (1–2 km distance). Blood samples were transported (at 4–8 °C) within 4 h of collection. Hb was read using HemoCue (HemoCue^®^ Hb 301, HemoCue AB, Ängelholm, Sweden), and G6PD activity (in U/g Hb) was measured using a point-of-care quantitative G6PD testing platform (SD Biosensor, Deogyeong-daero, Republic of Korea) [13], in accord with manufacturer’s instructions. Each sample was tested once, unless there was a ≥2 g/dL discordance of Hb reading between the device and HemoCue. Subjects with G6PD ≥ 6.0 U/g Hb (70% normal activity threshold) were enrolled, and the blood samples were transported for further serological testing at the Eijkman laboratory in Jakarta. Treatment was only given to those with positive serological results and/or having acute malaria. At the time when the case occurred, 518 of 622 screened schoolchildren had been enrolled, and 92 of the 104 excluded participants had G6PD activity values <6.0 U/g Hb. In all, 12 of the enrolled subjects had been treated with primaquine, whereas the remaining 506 were seronegative or in the control arm and thus were not treated.

The subject was a local resident screened for eligibility on the 22nd of February 2022. He was asymptomatic with a methaemoglobin (metHb) of 0.4% by non-invasive oximeter (Masimo, Irvine, CA, USA). His Hb was 12.0 g/dL, and G6PD was recorded as 6.3 U/g Hb. Treatment commenced a week following enrolment. The regimen consisted of three daily doses of dihydroartemisinin and piperaquine phosphate (DHP, one tablet = 40 mg/320 mg, respectively) and a daily dose of primaquine (PQ, 1 mg/kg BW, 15 mg tablet, PT Phapros, Jakarta, Indonesia) scheduled for 7 days. The subject (body weight 21.1 kg) was given 1.5 tablets of DHP and 1.25 tablets of PQ (18.75 mg; 0.89 mg/kg, Appendix A, Table A1), administered with a meal.

During the initial three days of treatment, there were no complaints or abnormal vital signs. On day 3, the subject’s Hb level dropped to 10.6 g/dL, a 12% decrease from baseline; his urine colour increased to a Hillmen score of 3 from a baseline of 2; and a third dose was administered. The following day, the subject experienced mild abdominal pain that rapidly resolved after a meal and antacid treatment. There was no further decline in Hb level (Hb = 10.8 g/dL) and a change in urine colour (Hillmen 4). The subject did not meet protocol criteria for treatment cessation (Hillmen >5 and fractional Hb fall ≥20% from baseline), and a fourth dose of PQ was administered.

On the following day (day 5), the subject’s Hb had dropped by 30.8% from baseline (from 12 g/dL to 8.3 g/dL), and his urine showed visible signs of haemolysis (Hillmen 5, Figure 1A). He appeared slightly weakened and complained of fever, headache, and abdominal pain but did not display clinical signs of jaundice or splenic enlargement. He had an elevated heart rate (116 beats/min) with a normal respiratory rate (20 breaths/min). The subject’s mother administered acetaminophen, and his temperature, measured during a home visit by the study physician two hours later, was 36.7 °C. The subject was admitted to the local primary health centre for supportive therapy, including IV fluid and oral paracetamol, and observation. No further primaquine dosing occurred. Appendix B, Table A5 lists clinical condition, laboratory observations, and relevant therapeutics from the initiation of therapy on day 1 to its cessation on day 5.

By day 6, the subject’s condition improved, and he showed more activity. There were no signs of jaundice, and his Hillmen score decreased from 4 in the morning to 2 in the afternoon (Figure 1B,C), although he remained visibly pale with a Hb level of 7.4 g/dL. Against the advice of the attending physician, his parents insisted on taking him home. The attending physician visited the boy at his home the next day (day 7), finding his condition alarmingly deteriorated: heart rate was 124 beats/min, and his haemoglobin had dropped to 5.4 g/dL. Although afebrile upon examination, he complained of fever since the prior day. The clinical team prevailed upon the parents to allow emergency transport to the provincial referral hospital in Medan, a 3 h drive. The subject was admitted to the hospital five hours later. Upon admission, the boy was fully conscious and had no difficulty with oral intake. He did not feel any dizziness upon sitting up and had normal temperature (37.0 °C) and heart rate (90 beats/min), but blood pressure was not checked. His sclerae were slightly icteric, and his conjunctivae were pale. There was no spleen enlargement. At admission, his blood showed a further decline of Hb to 4.7 g/dL (61% decrease from the pre-treatment value), with markedly elevated levels of reticulocytes and ferritin (Appendix C, Table A10). The overview of his clinical progression along with the daily Hb measurements and Hillmen score are illustrated in Figure 2.

The subject received a blood transfusion of 4 units (550 mL in total) of packed red cells starting on the first day of hospitalization. He remained in the hospital for four days, receiving supportive therapy, including supplemental oxygen and fresh frozen plasma due to a brief episode of upper gastrointestinal bleeding. He was discharged in good condition with a Hb level of 13.1 g/dL.

G6PD genotyping conducted the day after discharge (Appendix D) revealed a Coimbra variant. The field team visited the subject’s family to provide counselling regarding the diagnosis of G6PD deficiency with that particularly vulnerable variant. Three months after discharge, G6PD status confirmation at a certified laboratory employing quantitative spectrophotometry showed G6PD activity of 2.0 U/g Hb (normal reference: 7.8–14.4 U/g Hb).

A confirmed G6PD-deficient subject had been misclassified as normal. This likely occurred as a result of error in managing the sample in the laboratory during en masse G6PD screening. Investigation had shown the instrument used for screening to be in proper function and use by the operators. The clinical research team responded to the event and findings by implementing a second G6PD test at the point-of-care immediately prior to initiating PQ administration. No further similar events occurred.

### 2.2. Case 2

A 48-year-old man with *Plasmodium vivax* malaria was enrolled in a trial assessing the safety and efficacy of two primaquine dosing regimens (Artemether-Lumefantrin plus Primaquine vs. Dihydroartemisinin-piperaquine plus Primaquine) in the Solomon Islands (reg. no. ANZCTR 12617000329369). The trial was conducted from September 2017 to August 2019. A total of 629 individuals with *P. vivax* were screened, and 384 were enrolled. Ten of the excluded participants were identified as G6PD-deficient. Participants were recruited mainly by way of referral to the study team by clinical staff working in the local health facilities (passive case detection, PCD) or through active case detection (ACD) in the field. After informed consent was obtained, both CareStart™ Malaria and CareStart™ G6PD RDTs (RDT-1) were performed to confirm participant eligibility according to presence of *P. vivax* malaria and absence of G6PD deficiency, respectively. Baseline demographics and clinical characteristics were then collected, ensuring inclusion criteria were met. Participants were then transferred to the central study site for repeat G6PD screening with both CareStart™ and Binax Now™ RDTs (RDT-2 and RDT-3). Both RDT-2 and RDT-3 were performed under temperature-controlled conditions in line with the manufacturer’s recommendations. All three RDT results were checked and verified by a second operator at the time of testing. Only participants identified as G6PD-normal on all three RDT tests were enrolled.

The subject was enrolled on the 2nd of November 2017 after being cleared for inclusion. His body weight was 54 kg, his metHb by pulse oximetry (Masimo, Irvine, CA, USA) was 0.0%, and Hb estimation by HemoCue (HemoCue AB, Änglehom, Sweden) was 10.1 g/dL. He was randomized to receive primaquine (Primacin™, BNM Group, North Sydney, NSW, Australia) at a dose of 15 mg per day (according to a planned 14-day 0.25 mg/kg dosing schedule, Appendix A, Table A2), in combination with a 3-day course of artemether-lumefantrine. On the fourth day of treatment, after having received a total of 45 mg (0.75 mg/kg) of primaquine, he was noted by the study team to be pale, jaundiced, and complaining of fatigue. His Hb dropped to 7.9 g/dL, representing an 22% relative drop from a baseline value (10.1 g/dL). He had dark urine (Hillmen 5) that was strongly positive for urobilinogen on dipstick testing (8 E.U./dL). Although previously documented to not be G6PD-deficient (RDT1, 2 and 3), repeat testing with a BinaxNow™ kit on day 3 clearly indicated G6PD deficiency. No further primaquine was administered, and the participant made an uneventful recovery. The overview of the subject’s clinical progression as well as Hb and Hillmen score from the initiation of treatment to the onset of haemolysis are shown in Figure 2 and Appendix B, Table A6. On 21 May 2019, further quantitative G6PD with OSMMR2000-D G6PD Kit (R&D Diagnostics Ltd., Aghia Paraskevi, Greece) according to manufacturer’s instruction demonstrated G6PD activity of 7.06 U/g Hb. Sequencing of the G6PD gene by PacBio long-read amplicon sequencing [15] demonstrated a C-to-T substitution mutation at position 1360, consistent with a G6PD variant variously described as Union, Maewo, or Chinese-2.

### 2.3. Case 3

This was the second reported case in the trial conducted in the Solomon Islands (reg. no. ANZCTR 12617000329369). By the time both cases occurred, 72 participants had been enrolled in this trial. The subject was a 16-year-old male with mixed *P. vivax/P. falciparum* infection. His body weight was 42 kg, metHb was 0.4%, and Hb was 11.8 g/dL. He was enrolled on 19 December 2017 and randomized to receive primaquine (Primacin™, BNM Group, North Sydney, NSW, Australia) at a dose of 11.25 mg per day (according to a planned 14-day 0.25 mg/kg dosing schedule, Appendix A, Table A2), in combination with a standard 3-day course of artemether-lumefantrine. Routine review by the study team on days 1–4 of treatment documented normal urine colour (Hillmen 2 on day 4 of treatment) and no symptoms. However, at his next scheduled study visit on day 8, after having received a total of 79 mg (1.75 mg/kg) of primaquine, he complained of fever, nausea, and right upper quadrant pain over the previous 3 days (symptoms, therefore, commencing on day 5, at which time he would have received approximately 45 mg primaquine, Figure 2). He was noted by the study team to be pale, jaundiced, tachycardic (105 beats/min), and febrile (37.8 °C). Hb at the time was 6.0 g/dL, representing a 49% relative drop from a baseline Hb of 11.8 g/dL. He had dark urine (Hillmen 5) that was strongly positive for urobilinogen on dipstick testing (8 E.U./dL). Pulse oximetry (Masimo, Irvine, CA, USA) measured a methaemoglobin concentration of 1.7%. Again, although previously documented to have returned non-deficient screening results prior to treatment on point-of-care G6PD testing (RDTs1, 2, and 3 including both BinaxNow™ and CareStart™ tests), repeat testing with a BinaxNow™ kit on day 7 clearly indicated G6PD deficiency. The primaquine was stopped, and he was referred to the National Referral Hospital on the same day (day 8) to receive a blood transfusion.

Upon admission, his blood was taken for further laboratory testing. His blood slide was negative for malaria, and his Hb was 6.7 g/dL (moderate anaemia). He received one unit of red cells and ferrous sulphate supplementation and was discharged the next day (day 9).

The subject was subsequently reviewed on day 10, when his Hb was measured at 7.9 g/dL according to HemoCue. His respiratory rate was 20 breaths/min, and his heart rate was 62 beats/min. His metHb was 1.6%. By day 12, his Hb has increased to 10.3 g/dL with normal vital signs (respiratory rate = 22 breaths/min, heart rate = 87 beats/min), and metHb of 0.7%. He appeared well and had resumed daily regular activities.

Further quantitative testing with OSMMR2000-D G6PD Kit (R&D Diagnostics Ltd., Aghia Paraskevi, Greece) demonstrated G6PD activity of 1.45 U/g Hb (WHO Class B [16]). Sequencing of the G6PD gene [15] revealed the same C-to-T substitution mutation at position 1360, again consistent with the Union variant.

### 2.4. Case 4

A 20-year-old man was enrolled in a multicentre clinical trial to assess the noninferiority of a short duration of high-dose primaquine (7 days of 1 mg/kg/d vs. 14 days of 0.5 mg/kg/d) as a radical cure for *P. vivax* (IMPROV Study, reg no. NCT01814683). A total of 2336 participants from four countries (Afghanistan, Ethiopia, Indonesia, and Vietnam) were enrolled during the period of July 2014–November 2017 [17]. Subjects were recruited by PCD in a clinic located in Binh Phuoc Province, Vietnam. Those with clinical presentation of malaria, age ≥ 6 months, weight > 5 kg, Hb ≥ 9 g/dL, and normal G6PD were declared eligible for the study [17]. G6PD screening was performed on a venous blood sample by fluorescent spot test (FST; R&D Diagnostics, Athens, Greece) and CareStart™ RDT. Only those with normal results from both qualitative assays were enrolled in the study. Of the total 11,585 screened participants, 50 were identified as G6PD-deficient and excluded.

The subject (BW = 59 kg) was positive for *P. vivax* by light microscopy and had no other health issues. His Hb was 15.3 g/dL with normal G6PD screening results recorded. He was enrolled on the 16th of November 2014 and randomized to receive 7 days of daily 1 mg/kg BW primaquine (4 tablets of 15 mg, Centurion Laboratories, Vadodara, India, Appendix A, Table A3). Chloroquine was given as the blood schizontocide, according to national guidance (5 doses of 2 tabs of 250 mg).

On day 2, after receiving his first dose of primaquine (1 mg/kg), his urine Hillmen increased two levels from baseline (2 to 4), but this was not accompanied by any complaint or physical abnormalities. On day 3, however, after having received two doses of primaquine (2 mg/kg), his Hb had dropped 21% from baseline to 12.1 g/dL. No further change was observed in his urine colour, and a third PQ dose was administered. On day 4, his Hb further decreased to 11.5 g/dL, but because he was asymptomatic with a stable Hillmen score at 4, he was given another PQ dose. On day 5, his Hb fell to 9.7 g/dL (37% drop from baseline), but his urine colour at Hillmen 4–5 remained stable. His temperature was 37.5 °C with a heart rate of 84 beats/min and a blood pressure of 110/70 mm Hg. The patient was given a fifth PQ dose despite the presence of cessation criteria (fractional Hb fall ≥25% from baseline). Clinical signs of acute haemolysis started to be noticeable on the following day (day 6), when his Hb level decreased to 7.6 g/dL, a 50% drop from baseline value; he was pale and slightly jaundiced with dark urine (Hillmen 5). He complained of abdominal pain and was feverish (37.6 °C) and tachycardic (pulse = 96 beats/min). His blood pressure was 120/80 mm Hg, and his respiratory rate was 24 breaths/min. Details on his clinical progression from baseline to the onset of the haemolytic episode are described in Appendix B, Table A8. PQ was stopped, and the patient was admitted to the hospital for close observation of his vital signs and further laboratory workup (Appendix C, Table A10).

On the following day (day 8), his Hb decreased further to 6.6 g/dL. He had no fever, and his urine colour had lightened (Hillmen 3–4). His Hb dropped slightly further to 6.4 g/dL the day after (day 9), as shown in Figure 2. The G6PD test was repeated by FST, and he was found to be G6PD-deficient. He was transfused (250 mL of packed red cells), and his Hb rose to 8.0 g/dL (day 10), which slightly dropped to 7.7 g/dL on the following day (day 11). Two days later (day 12), his Hb had increased to 9.4 g/dL, and he was discharged from the hospital. His blood was genotyped and found to be a Viangchan variant.

### 2.5. Case 5

The subject was a local 13-year-old boy enrolled in a trial assessing the efficacy of spatial repellent to reduce malaria transmission in Southwest Sumba, Indonesia during September 2012–April 2013 [18]. In this trial, 180 G6PD-normal children aged 6–13 years were recruited for presumptive radical cure prior to a 26-week follow-up period as a means of measuring the incidence density of new malaria infections in the community. Following informed consent and demographic data collection, all participants were examined by the study physician, body weights were measured, and haemoglobin concentrations were measured by HemoCue^®^ Hb 201+ (HemoCue AB, Ängelholm, Sweden). First, 3 mL of venous blood was collected in EDTA tubes, stored temporarily in a cooler box, and transported to a site laboratory located in a town 69 km away within 4 h of collection. The laboratory performed G6PD FST (Trinity Biotech qualitative G6PD assay^TM^, ref 345-UV, Trinity Biotech, St. Louis, MO, USA). The results were conveyed to the field team within 24 h via mobile phone text message. A total of 25 of the 231 screened participants were identified as G6PD-deficient and excluded.

The boy was screened on 18 October 2012. His weight was 29 kg, and his Hb was 11.3 g/dL. His G6PD status was recorded as normal, and he was enrolled in the study. Therapy with standard dihydroartemisinin-piperaquine (DHP) was initiated immediately, followed by primaquine the next day. The regimen for DHP was two tablets for three days, and the primaquine dose was 0.6 mg/kg (1.25 tab, 15 mg tablet, Kimia Farma, Semarang, Indonesia) for 14 days. On day 3 after receiving two doses of primaquine (approximately 1.2 mg/kg), he complained of nausea with epigastric pain but did not appear anaemic or jaundiced. His Hb was not measured, and urine colour was not checked. A third dose of primaquine was given (total 1.8 mg/kg). On the following day, he complained again of nausea and fever, but his temperature on examination was 36.4 °C. The site physician then gave him ranitidine and metoclopramide. As the complaints subsided, a fourth dose of primaquine was administered (total 2.4 mg/kg). The next day, the patient visited the clinic with nausea and vomiting. He was also pale, icteric, and tachycardic. His spleen was enlarged on physical examination. His urine was dark, and his Hb level was 7.2 g/dL. His clinical progression and Hb drop are summarized in Figure 2 and Appendix B, Table A9. Primaquine was stopped, and the boy was referred to the nearest hospital for further management. His FST was repeated using Trinity Biotech qualitative G6PD assay™ (Trinity Biotech, St. Louis, MO, USA) and showed a normal G6PD result.

At the hospital (6 h after cessation of primaquine), his Hb dropped further to 5.6 g/dL (50% from his baseline value, Figure 2), with markedly elevated white blood cells (23,600/µL vs. 4500–13,500/µL) and bilirubin (total: 8.8 mg/dL vs. 0.2–1.2 mg/dL, direct: 0.7 mg/dL vs. <0.5 mg/dL). His urea was slightly elevated (66 mg/dL vs. 19–44 mg/dL), but his creatinine was normal, consistent with dehydration (Appendix C, Table A10). His urine was tea-coloured and strongly indicated the presence of urobilinogen and red blood cells (urine dipstick: blood +3 and urobilinogen 8 E.U/dL or 131 µmol/L). He received a blood transfusion the following day with a total of 500 mL of packed red cells. He was discharged after seven days of hospitalization with an Hb of 10.7 g/dL. His Hb has returned to normal (12.2 g/dL) during a follow-up visit 11 days after discharge. The G6PD test was repeated three months later using the Trinity Biotech qualitative FST G6PD assay™ (Trinity Biotech, St. Louis, MO, USA), with findings consistent with G6PD deficiency. The genotyping result showed Vanua Lava variant, which turned out to be the dominant G6PD variant in this population [9].

## 3. Discussion

We observed five episodes of acute haemolytic anaemia detected between days 4 and 8 during the course of treatment with daily primaquine anti-relapse therapy. The patients were enrolled participants in four separate RCTs and received a range of daily primaquine doses (0.25–1.00 mg/kg) after seemingly testing G6PD-normal with qualitative or quantitative screening tests. All events occurred in the setting of rigorous clinical trial processes and procedures, including well-defined ascertainment strategies, careful screening protocols, close monitoring, and well-prepared mitigation plans in the event of inadvertent treatment of G6PD-deficient subjects due to screening failure. All of these patients fully recovered, four of them after blood transfusion therapy.

The five cases described here document the clinical implications of what clearly represents failures of screening processes intended to protect G6PD-deficient patients from exposure to 8-aminoquinoline therapy. The precise reasons underlying each of these failures are difficult to discern. However, it is notable that they occurred in four separate clinical trials in three different countries. All employed high standards of clinical practice and oversight, carefully complying with protocols designed to minimize haemolytic risk. It is particularly important to note that the four trials utilized a variety of testing platforms, with a total of five different tests used including the SD biosensor (Case 1), CareStart™ RDT (Cases 2–4), Binax now RDT (Cases 2 and 3), FST R&D Diagnostics (Case 4), and FST Trinity Biotech qualitative test (Case 5). This points to two important possible factors at play here. Firstly, no G6PD field test can be assumed to have 100% sensitivity (and therefore flawless negative predictive value) even under ideal laboratory operating conditions [19,20,21,22]. However, currently available point-of-care G6PD tests represent evolving technologies, and improved sensitivity and specificity may yet be achieved [23]. Secondly, a number of operational factors may compromise diagnostic accuracy when deployed in field conditions. These could potentially include operator errors in sample preparation and processing, cold chain issues compromising sample or screening kit integrity, lack of adequate climate control to ensure tests are conducted at appropriate temperature conditions, and operator errors in reading and interpreting tests (e.g., in operators with reduced visual acuity, physically exhausted, or overwhelming workload and rush). Taken together with the inherent limitations of testing platforms, these all suggest that whilst we can do our best to minimize the risk of screening failure, it seems unlikely we are able to eliminate this risk altogether. It is particularly notable that in one trial, screening failure occurred in spite of triplicate RDT testing with two separate tests and dual operator verification. Therefore, whilst rigorous training in G6PD test application will be required when primaquine and tafenoquine are rolled out, we should still expect to see screening failures and have systems in place for early detection and clinical rescue of acute haemolytic events. Indeed, these systems were in place as part of the protocolized safety mechanisms used in each of the four clinical trials. These very likely prevented much more serious events, including renal failure and death. We report these experiences in order to highlight the vital importance of including similar failsafe countermeasures in deployment strategies for 8-aminoquinoline anti-relapse treatments.

The cardinal laboratory sign of acute haemolytic anaemia is a “steep” drop in measured Hb levels in blood that may be accompanied by symptoms of anaemia and/or signs like dark urine, pale conjunctivae, and yellow sclera. In three cases, these signs were not detected early enough to prompt a decision to stop PQ, whilst in the other two patients, the clinical course may have been different if PQ had been stopped earlier, as mandated by the protocol (Case 4), or had extra vigilance been exercised by re-assessing the Hb concentration and urine colour (Case 5). In the four cases with available daily Hb data, the course of haemolysis appeared only slightly after the first two doses; Hb dropped only by 10–20%, well within the normal range for a patient undergoing antimalarial therapy [24]. Excepting only one patient (Case 2), Hb levels remained relatively unchanged after the third dose. This pattern was also seen in the African A- variant and is consistent with reduced glutathione being depleted as it buffers the primaquine-induced oxidant stress [25]. It was only 24 h after the fourth dose (day 5) that Hb levels commenced steep declines with fractional drops of 31–37% from baseline along with notably dark urine (Hillmen 4 or 5). In the only patient without daily Hb reading after the fourth dose (Case 3), the Hb had fallen to 49% of baseline value (11.8 to 6.0 g/dL) on day 8 after the seventh dose.

Clinical features in the days leading up to the haemolytic crisis were not specific, as the common symptoms (nausea and fever) often occur in the days following treatment for acute malaria, and abdominal discomfort is a common complaint from patients consuming PQ anti-relapse therapy, more so with chloroquine compared to dihydroartemisinin piperaquine [26]. The MetHb level (measured in Cases 1–3) was not predictive either, and only Case 2 showed an extraordinary elevation of metHb (to 15.8%). In all four subjects thus examined on the day of onset of the haemolytic crisis, urine urobilinogen was conspicuously elevated (10- to 25-fold above the baseline of 0.2 E.U./dL). The daily checks of this measurement in Case 1 showed an early 5-fold elevation of urobilinogen without the detection of urine bilirubin (the typical finding of prehepatic jaundice) on day 2 despite a low (<10%) fall in Hb. The early detection of increased urobilinogen alone may be the earliest sign of incipient severe haemolysis and may have greater predictive value than dark urine, which is caused by free haemoglobin in urine consequent to intravascular haemolysis. This may as well be observed in continuing increase of Hillmen score, which was recorded in two cases (Cases 1 and 4). This hypothesis will require validation in more patients.

The failure to detect haemolysis early will result in continuing haemolytic crisis with continued exposure to PQ dosing. This may be particularly dangerous when the G6PD deficiency variant involved is of a severely impaired enzyme activity phenotype. All of the four known variants involved in five cases in this series—Coimbra, Union, Viangchan, and Vanua Lava—were of that phenotype. Under the new WHO classification of G6PD variants, these are all Class B, with residual enzymatic activity of <45% of normal [16]. Coimbra and Union variants in particular are among those resembling the phenotype of the Mediterranean variant, with severely impaired G6PD activity even in reticulocytes [27], which may not apply to the Viangchan variant [28]. Interestingly, the G6PD activity was unexpectedly high in Case 2 given the presence of Union variant and clinically significant haemolysis. We do not have any obvious explanation for this.

In two of the cases reported here (Cases 1 and 4), the haemolytic crisis did not halt with the cessation of dosing but continued for the 2–3 days leading up to hospitalization and transfusion. This has been observed by others [29,30,31]. In the instance of Case 5, the crisis progressed so rapidly that transfusion occurred on the same day of PQ cessation (Figure 2). These patients kept deteriorating even after PQ dosing had ceased. This ongoing haemolysis is probably due to continuing splenic extravascular haemolysis of oxidant-damaged red blood cells [27], although Case 1 had evidence of upper gastrointestinal bleeding, and Chu et al. [29] suggested a concomitant bacterial infection in one patient and vitamin C administration in another.

The haemolytic events reported here highlight the danger daily primaquine poses to patients having G6PD deficiency; this may be equally or more challenging with single-dose tafenoquine anti-relapse therapy, a drug having a mean half-life of ~14 days [32] and thus prolonged haemolytic potential. The new WHO classification scheme no longer distinguishes the former Class II and Class III variants, represented by Mediterranean and African A- archetypes of so-called “severe” and “mild” variants. These have been reclassified as type B variants in acknowledging all variants as potentially dangerous to patients who receive haemolytic drugs. However, important physiological distinctions remain; i.e., the Mediterranean phenotype does not develop tolerance of primaquine dosing with the onset of reticulocytosis, whereas the A- phenotype does [25,33,34,35].

The experiences reported here demonstrate the importance of escalated vigilance (follow-up and patient education), particularly during days 3–7 of PQ administration, even when G6PD screening is implemented. This principle is exemplified in Case 4, where the decision to continue with primaquine despite a notable decline in haemoglobin levels was underscored by the patient’s apparently healthy appearance and stable Hillmen colour. In retrospect, primaquine should have been stopped earlier, even if the declining haemoglobin level was consistent with malaria. Close monitoring, interrupting primaquine administration upon any signs of haemolysis, and resuming treatment only when considered safe emerge as pivotal measures when administering primaquine. More research is needed to identify early markers that may predict with confidence the early onset of a severe haemolytic crisis with continued primaquine exposure, e.g., clinical symptoms with increasing Hillmen/urine urobilinogen. The onset of dark urine was clearly a late sign in our series but has also been seen with accidental primaquine overdose and in favism, in which the acute development of dark urine was associated with acute catastrophic falls in haemoglobin [29,36,37].

## 4. Conclusions

Primaquine poses the potential danger of severe haemolysis in individuals with G6PD deficiency. The drug should be given with extra vigilance, especially during days 3–7 of treatment. Screening for G6PD deficiency is imperative before initiating the treatment, and there should be a robust plan in place to promptly rescue patients from a haemolytic crisis, should it arise.

## Figures and Tables

**Figure 1 pathogens-12-01176-f001:**
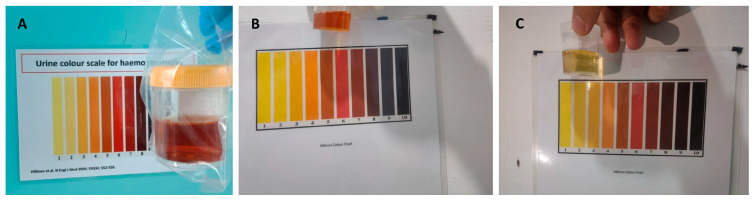
Subject’s urine macroscopic result. Dark urine colour suggesting haemolysis (Hillmen scale = 5) was shown on the day 5 of treatment (**A**). On the following day (day 6 of treatment), the colour was less dark (Hillmen scale = 4 and 2 in the morning (**B**) and afternoon (**C**), respectively).

**Figure 2 pathogens-12-01176-f002:**
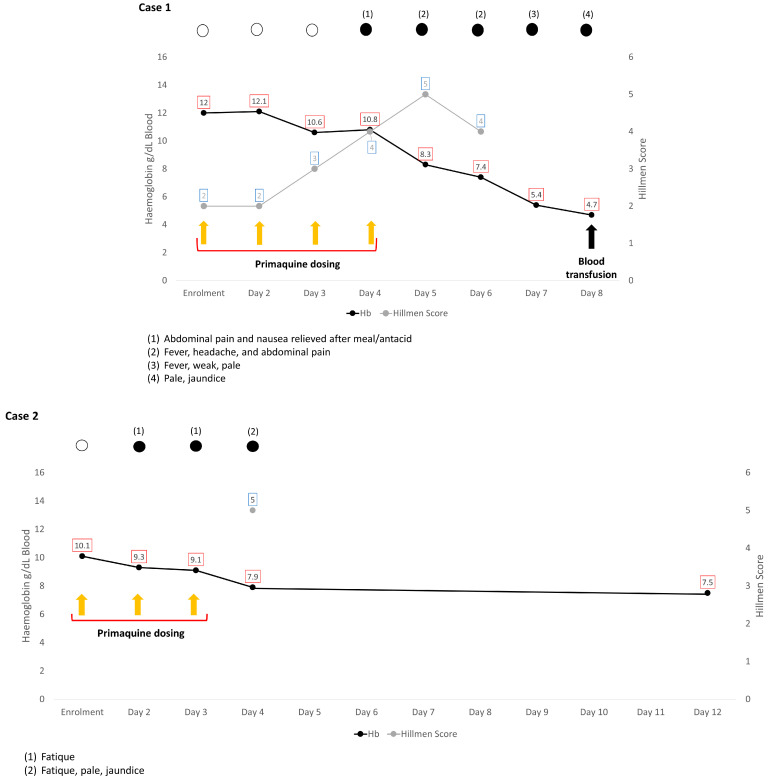
Graphical display of the clinical progression and temporal changes of haemoglobin and urine Hillmen score during primaquine administration and 1–8 days following its cessation. In all cases but one, clinical symptoms had been observed 1–3 days before the onset, along with the slight decline of Hb. Of note, steady increase of Hillmen as the Hb dropped was observed in **Case 1** and **4**, in which daily urine inspection were performed. In all cases, primaquine was interrupted at the same day of the onset of the haemolysis. In **Case 1** and **4**, the crisis continued with further blood loss leading to transfusion. In **Case 3** and **5**, the anaemia was severe (Hb 5–6 g/dL); the patients were given transfusion promptly after haemolysis was suspected. **Case 2** became the only case where blood transfusion was not indicated.

## Data Availability

The data presented in this study are available on request from the corresponding author. The data are not publicly available due to the privacy protection of the subjects presented here.

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
