# Peer review of "Case Series of Primaquine-Induced Haemolytic Events in Controlled Trials with G6PD Screening"

_pathogens, 2023, doi:10.3390/pathogens12091176_

Round 1
Reviewer 1 Report
In this article the authors describe 5 cases of severe haemolytic anaemia that occurred in participants enrolled in RCTs and administered primaquine, and subsequently confirmed to be G6PD deficient. These events occurred in 4 RCTs conducted during 2012 – 2022.
Major comments:
- The authors do not describe their methods. Ie, how were these cases identified? How many RCTs were reviewed to identify these cases? How were the RCTs reviewed?
- - Given the number of participants enrolled in RCTs during the 5 year period, these 5 cases presumably represent a very small proportion. It would be helpful for the authors to provide some estimates of the denominator – ie, how many patients in total were enrolled in these 4 trials? And how many participants were enrolled in other RCTs conducted during this time, that involved administration of PQ, and didn’t result in any severe haemolytic events?
- - Overall, the description of the cases is very wordy and would benefit from some editing to make the writing more concise.
Other comments:
- - Abstract, line 35. The sentence: “signs of acute haemolysis, ie., dark urine and Hb drop >20%, were not observed in any patient before day 4 of treatment and as late as day 8” – is hard to understand. Suggest reword to describe the events and when they occurred, rather than when they did not occur.
- - Figure 2 is too small to read.
- - Case 2 had G6PD activity of 7.06 U/g Hb. Could the authors comment on the development of haemolysis in this patient, despite this level of G6PD activity?
- - Case 4 – the patient continued to receive PQ despite concerning falling Hb and an increase in the urine Hillmen score (particularly on day 5, when PQ was given despite a Hb drop of 37% from baseline). Was this against the study protocol? The authors could perhaps mention that this particular case represents a failure of clinical monitoring/management rather than just a failure of the G6PD screening.
- - Line 317 - 319 – no units provided for white blood cell count or urea. Also, the authors state that “urea was slightly elevated (66 vs 19-44)”, but it is not clear what these values represent. Same for white cell count and bilirubin.
- - In the discussion section the authors make several references to the rigorous monitoring and systems in place to detect and manage haemolytic evens. However, as mentioned above, in Case 4 the patient continued to receive PQ despite a concerning drop in Hb. So these “failsafe countermeasures” (line 369) are perhaps not quite as “failsafe” as suggested by the authors.
- - The authors discuss the importance of close monitoring as occurred in these trials. However, in reality, this degree of monitoring will not occur outside of clinical trials (ie, daily Hb or physical examination). Could the authors comment on the specific monitoring that should occur outside of clinical trials?
- - The paper would benefit from a couple of concluding sentences.
A few grammatical errors throughout, and could do with some editing to improve the language. Also the case descriptions are rather wordy, and would benefit from some editing to make the language more concise.
Author Response
Reviewer #1:
- The authors do not describe their methods. Ie, how were these cases identified? How many RCTs were reviewed to identify these cases? How were the RCTs reviewed?
Response: This was not a systematic review so there was no exhaustive literature search. All patient data were given to us by colleagues and collaborators. We have added the information on page 2, line 85-86.
“The authors of this report are colleagues and collaborators who are aware of each other’s work and experiences.”
- Given the number of participants enrolled in RCTs during the 5 year period, these 5 cases presumably represent a very small proportion. It would be helpful for the authors to provide some estimates of the denominator – ie, how many patients in total were enrolled in these 4 trials? And how many participants were enrolled in other RCTs conducted during this time, that involved administration of PQ, and didn’t result in any severe haemolytic events?
Response: Thank you for this valuable feedback. We have revised the paper accordingly. We also added information on the number of participants identified with G6PD deficiency.
Case #1 (Line 113-115):
“At the time when the case occurred, 518 of 622 screened schoolchildren had been enrolled, and 92 of 104 excluded participants had G6PD < 6.0 U/g Hb. Twelve of the enrolled subjects had been treated with primaquine.”
Case #2 and #3
“Ten of excluded participants were identified as G6PD deficient.” (Line 202)
“By the time both cases occurred, 72 participants had been enrolled in this trial.” (Line 237)
Case #4
“A total of 2336 participants from four countries (Afghanistan, Ethiopia, Indonesia, and Vietnam) were enrolled during the period of July 2014-November 2017 [18] (Line 274-276)
“Of the total 11,585 screened participants, 50 were identified as G6PD deficient and excluded.” (Line 282-283)
Case #5
“The subject was a local 13-year-old boy enrolled in a trial assessing the efficacy of spatial repellent to reduce malaria transmission in Southwest Sumba, Indonesia during September 2012-April 2013 [15]. In this trial, 180 G6PD-normal children aged 6-13 years were recruited for presumptive radical cure prior to a 26-week follow-up period as a means of measuring the incidence density of new malaria infections in the community.“ (316-320)
“Twenty-five of 231 screened participants were identified as G6PD deficient and excluded.” (Line 329)
As previously mentioned, our intention was not a systematic review and estimation of the population risk of primaquine induced acute haemolytic anaemia. Therefore, we do not know of similar events occurring in other RCTs during the same period. Nevertheless, evidence to date suggests the incident of G6PD screening failure in the research setting leading to severe hemolytic reactions appears rare. Our paper highlights this risk to raise awareness of the potential implications when implementing radical cure in the community to treat patients or as mass drug administration.
- Overall, the description of the cases is very wordy and would benefit from some editing to make the writing more concise.
Response: Thank you for your input. We have made some edits to it.
“One week following enrolment, serological test showed a positive result and treatment commenced the following day” has changed to “Treatment commenced a week following enrolment.” (Line 118-119)
“On Day 3, the subject’s Hb level slightly dropped to 10.6 g/dL (a 12% decrease from baseline), accompanied by a slight change in urine colour” has changed to “On Day 3, the subject’s Hb level slightly dropped to 10.6 g/dL (a 12% decrease from baseline), and urine colour slightly changed” (Line 126-127)
“Against the advice of the attending physician, his parents insisted on taking him home, and he was discharged from the health centre” has changed to “Against the advice of the attending physician, his parents insisted on taking him home.” (Line 151-152)
“At admission, his blood showed a further decline of Hb to 4.7 g/dL from the last examined level of 5.4 g/dL” has changed to “At admission, his blood showed a further decline of Hb to 4.7 g/dL” (Line 161)
“His G6PD screening results were recorded as normal” has changed to “His G6PD status was recorded as normal…” (Line 331)
“Therapy with standard dihydroartemisinin-piperaquine (DHP) was initiated immediately and primaquine was initiated on the following day” has changed to “Therapy with standard dihydroartemisinin-piperaquine (DHP) was initiated immediately, followed by primaquine the next day.” (Line 331-333)
- Abstract, line 35. The sentence: “signs of acute haemolysis, ie., dark urine and Hb drop >20%, were not observed in any patient before day 4 of treatment and as late as day 8” – is hard to understand. Suggest reword to describe the events and when they occurred, rather than when they did not occur.
Response: Thank you for your input. We have revised it (Line 35-36).
“Early signs of acute haemolysis, i.e., dark urine and haemoglobin drop >20%, occurred only after day 3 and as late as day 8 of primaquine dosing.”
- Figure 2 is too small to read.
Response: Thank you for your comment. We have improved it (page 4-6)
- Case 2 had G6PD activity of 7.06 U/g Hb. Could the authors comment on the development of haemolysis in this patient, despite this level of G6PD activity?
Response: We agree with the reviewer that the relatively high enzyme activity documented in Case 2 is perplexing and difficult to explain. Samples were assayed by well-trained laboratory technician with duplicates sample and both come up with similar interpretation (7.06 and 5.51 U/g Hb or 59% and 46% activity compared to normal control of11.97 U/g Hb). Samples were run some time after the initial primaquine induced haemolysis episode described, raising questions about intercurrent factors at the time of testing (eg unknown recent episode of haemolysis or transfusion) that may have lead to a spurious result. But this would be purely speculative. We think it is best to simply report our objective findings and let the reader come to their own conclusions. We have added a note to the manuscript as follows (Line 444-446)
“Interestingly, the G6PD activity was unexpectedly high in Case 2 given the presence of Union variant and clinically significant haemolysis. We don’t have any obvious explanation for this.”
- Case 4 – the patient continued to receive PQ despite concerning falling Hb and an increase in the urine Hillmen score (particularly on day 5, when PQ was given despite a Hb drop of 37% from baseline). Was this against the study protocol? The authors could perhaps mention that this particular case represents a failure of clinical monitoring/management rather than just a failure of the G6PD screening.
Response: Thank you for the constructive feedback. The study SOP stipulated a clinical review and withholding primaquine if there was a fractional fall in haemoglobin (Hb) of >25%. Despite this, primaquine was continued based on clinical judgement: he was well despite a falling Hb, and although his Hillmen score did increase it remained stable, perhaps suggesting poor sensitivity. Only when there were overt signs of haemolysis was his primaquine stopped.
In retrospect, we would agree that the management of this patient was less than optimal and his primaquine should have been stopped earlier even if the clinicians thought initially that the falling Hb was consistent with malaria itself.
This is a key message of our paper – close monitoring, interrupt primaquine, and resume treatment when considered safe to do so.
In the discussion we have added (Line 468-475):
“This principle is exemplified in Case 4, where the decision to continue with primaquine despite a notable decline in haemoglobin levels is underscored by the patient's apparently healthy appearance and stable Hillmen color. In retrospect, primaquine should have been stopped earlier, even if the declining haemoglobin level was consistent with malaria. Close monitoring, interrupting primaquine administration upon any signs of haemolysis, and resuming treatment only when considered safe, emerge as pivotal measures when administering primaquine.”
- Line 317 - 319 – no units provided for white blood cell count or urea. Also, the authors state that “urea was slightly elevated (66 vs 19-44)”, but it is not clear what these values represent. Same for white cell count and bilirubin.
Response: Thank you for this feedback. We have revised it accordingly (Line 348-350)
“…..markedly elevated white blood cells (23,600/µL / vs. 4500-13,500/µL) and bilirubin (total: 8.8 mg/dL vs. 0.2-1.2 mg/dL, direct: 0.7 mg/dL vs. < 0.5 mg/dL). His urea was slightly elevated (66 mg/dL vs 19-44 mg/dL)….”
- In the discussion section the authors make several references to the rigorous monitoring and systems in place to detect and manage haemolytic evens. However, as mentioned above, in Case 4 the patient continued to receive PQ despite a concerning drop in Hb. So these “failsafe countermeasures” (line 369) are perhaps not quite as “failsafe” as suggested by the authors.
Respnse: Thank you for your input. As we mentioned earlier, we have addressed this particular situation as less than optimal patient management (please refer to no 7).
- The authors discuss the importance of close monitoring as occurred in these trials. However, in reality, this degree of monitoring will not occur outside of clinical trials (ie, daily Hb or physical examination). Could the authors comment on the specific monitoring that should occur outside of clinical trials?
Response: Thank you for this critical insight. We write this report to send a message for programmer or healthcare provider regarding the potential danger of this drug and that current monitoring practice is not enough to prevent catastrophic event led by haemolysis. Given the limited number of the cases presented in this report, we cannot give any recommendation to provide a safer way to administer the drug. Providing daily monitoring of clinical symptoms and Hillmen/urine urobilinogen may be of benefit, but this requires more research for validation of its use (Line 474-477).
“More research is needed to identify early markers that may predict with confidence the early onset of a severe haemolytic crisis with continued primaquine exposure, e.g. clinical symptoms with increasing Hillmen/urine urobilinogen.“
- The paper would benefit from a couple of concluding sentences.
Response: Thank you for this useful comment. We have added the conclusion section (Line 482-87)
“Primaquine poses potential danger of severe hemolysis in individuals with G6PD deficiency. The drug should be given with extra vigilance, especially during days 3-7 of treatment. Screening for G6PD deficiency is imperative before initiating primaquine, and there should be a robust plan in place to promptly rescue patients from a hemolytic crisis, should it arise.”
- A few grammatical errors throughout, and could do with some editing to improve the language. Also the case descriptions are rather wordy, and would benefit from some editing to make the language more concise.
Response: Thank you for your input. We have made some grammar corrections to the paper.
“The safe application of primaquine anti-relapse therapy requires G6PD screening and anticipation of its failure with a means of prompt detection and rescue from the typically abrupt haemolytic crisis.” (Line 42)
“Healthcare providers managing P. vivax patients of unknown G6PD status must choose between a therapy inviting the risk of acute haemolytic anaemia” (Line 63)
“Physical examination by the study physician and capillary blood collection from finger prick were was done on-site,..” (Line 104)
“A 20-year-old man was enrolled in a multicentremulticenter clinical trial..” (Line 272)
“…his Hb level decreased to 7.6 g/dL, a 50% drop from baseline value; he was pale and slightly jaundice jaundiced with dark urine…” (Line 301)
“Details on his clinical progression from baseline to the onset of the haemolytic episode is are described in SupplementaryTable B4.” (Line 303-305)
“In the four cases with available daily Hb data, the course of haemolysis appeared only slight slightly after the first two doses;..” (Line 410)
We also have made some edits to the cases (please refer to no.3)
Reviewer 2 Report
Pathogens-2527277-peer-review-v1-comments
Case Series
Case Series of Primaquine-Induced Haemolytic Events in Controlled Trials with G6PD Screening
Ayleen Kosasih et.al.
Overall Comments :
Radical cure for patients suffering from Plasmodium vivax and Plasmodium ovale malaria is important for the patient as well as the community. To achieve an effective radical cure, a sufficient total dose of PQ needs to be administered over a prolonged course, which is usually 14 days for G6PD-normal patients and weekly for 8 weeks for G6PD-deficient patients. Recently Tafenoquine has also been considered as the most promising drug for clearing the liver stages(Hypnozoites) but G6PD testing before administration is essential. G6PD deficiency is known to affect a large population of more than approximately 400 million people worldwide and the majority of them are at risk for malaria. Considering these important facts, the paper entitled, ‘ Case Series of Primaquine-Induced Haemolytic Events in Controlled Trials with G6PD Screening’ by Ayleen Kosasih et.al. is an important contribution to the existing literature.
The paper presents 5 events of acute haemolytic anaemia following the administration of primaquine in four malaria trials from Indonesia, the Solomon Islands, and Vietnam have been reported in the paper. These five males aged 9 to 48 years, were included in trials of anti-relapse therapy with daily doses of primaquine after testing and classifying them as G6PD-normal by various screening procedures. However, due to improper testing, these five males showed signs of acute haemolysis, i.e., dark urine and haemoglobin drop >20%, not before 4 days of PQ administration and late as day 8. Since the patients were a part of the supervised trial, all patients received the required medical care after hospitalization and recovered fully. The study points out the need for supervised PQ treatment in all settings in anticipation of its failure means of prompt detection and rescue from the typically abrupt haemolytic crisis, despite G6PD screening and classification of the patients as G6PD normal. The study also points out the doubtful efficacy of the available G6PD tests and the lack of adequate skills of the testing personnel leading to false G6PD normal results. From a public health point of view, this paper shows that Primaquine radical cure for the control and elimination of vivax malaria necessitates its safe delivery and building appropriate capacity to test G6PD status.
Overall, the paper is well written and the clinical details are well presented. However, because this paper has a bearing towards refining strategies and policy for future testing and treatment of malaria patients harbouring relapsing malarial patients, a few suggestions to improve the content are given below under specific comments
Specific comments:
Title & Abstract: The title is fine and the abstract conveys the crux of this paper adequately.
Introduction: The introduction is good but I recommend that it should be refined further to include updated information on the global burden of P.vivax malaria, G-6PD deficiency including brief data on the prevalence of G-6PD deficiency, if any, in the trial locations, recommended testing and treatment strategies of WHO as well as the latest G6PD testing options. The trials under which these five patients were included also need a little more introduction and elaboration for a better understanding of the background of these trials and their objectives for the readers. It would also be feasible to know the brief methodology and important brief details about the total number of subjects enrolled in each trial from the study areas i.e., Indonesia, the Solomon Islands, and Vietnam, of which these 5 patients suffered the adverse effects. Please also mention the period over which these trials were conducted and what was the year of study. These details will add to the importance of this paper from a public health perspective and at the same time provide a better perspective for understanding the background to the readers.
Case Series 1-5: The case series reports mention the years 2012 to 2022. Please mention clearly with references in the introductory lines of each case series whether the findings of these case reports have been published as a part of the trials elsewhere also previously or not.
Several tables have been referred to in the case series but none could be seen in the paper. Please check this and clarify.
Figure 2 is blurred and not legible. Please replace this with a clear and legible figure.
Abbreviations: It is better to write the complete form followed by abbreviations in parenthesis in the text instead of presenting them under the heading ‘Conflict of interest’. A separate table is not required.
Discussion :
The G6PD testing methods i.e., SD biosensor (Case 1), CareStart™ RDT (Cases 2, 3, and 4), Binax now RDT (Cases 2 and 3), FST R&D Diagnostics (Case 4) and FST Trinity Biotech qualitative test (Case 5)have undergone a lot of refinement during the recent years after 2012. Please ensure that the discussion considers these facts and limitations and that future suggestions are made accordingly keeping the updated treatment and testing strategies and options in mind.
References :
Please ensure that all references are as per the journal format.

Author Response
Reviewer 2
- Introduction: The introduction is good but I recommend that it should be refined further to include updated information on the global burden of P.vivax malaria, G-6PD deficiency including brief data on the prevalence of G-6PD deficiency, if any, in the trial locations, recommended testing and treatment strategies of WHO as well as the latest G6PD testing options. The trials under which these five patients were included also need a little more introduction and elaboration for a better understanding of the background of these trials and their objectives for the readers. It would also be feasible to know the brief methodology and important brief details about the total number of subjects enrolled in each trial from the study areas i.e., Indonesia, the Solomon Islands, and Vietnam, of which these 5 patients suffered the adverse effects. Please also mention the period over which these trials were conducted and what was the year of study. These details will add to the importance of this paper from a public health perspective and at the same time provide a better perspective for understanding the background to the readers.
Response: Thank you for the insightful comment. We have added more information in the introduction as well as in each case description.
Introduction
Line 48-54:
“Plasmodium vivax is widespread in many regions of the world and is particularly prevalent in countries in South Asia, Southeast Asia, Oceania, and parts of South America [1]. It is estimated that this species caused 4.9 million cases in 2021 globally [1]. Its ability to form a dormant stage in the liver poses great challenge to detect and target this species for intervention [2]. Infections arising from re-awakened hypnozoites are deemed to be the most problematic for the elimination program [2]. It has been estimated that 66%-96% of all P. vivax blood-stage infections are due to relapses [3].”
Line 59-62:
“….affecting an average of 8% of residents of malaria-endemic countries, with prevalence varying from 1-35% [7]. In Indonesia, the prevalence was reported to be 2.6% in North Sumatra [8] and 3.8% in Sumba [9]. In Vietnam, the prevalence was 8.9% [10]. In the Solomon Islands, the prevalence was 20.3% [11].”
Line 77-79:
“WHO strongly recommends the examination of G6PD status before administering primaquine as a radical cure for Plasmodium vivax [4].”
Case Description
Case 1 (Line 98-101):
“A total of 1133 participants aged 6–15-year-old were enrolled from February 2022-May 2023. Serological screening was performed to predict recent P. vivax exposure which may indicate the risk to harbour hypnozoites for which primaquine was given as a radical cure to prevent future relapses.”
Case 2 & 3
Line 197-201:
“A 48 year-old man with Plasmodium vivax malaria was enrolled in a trial assessing the safety and efficacy of two primaquine dosing regimens (Artemether-Lumefantrin plus Primaquine vs. Dihydroartemisinin-piperaquine plus Primaquine) in the Solomon Islands (reg. no. ANZCTR 12617000329369). The trial was conducted from September 2017 to August 2019. A total of 629 individuals with P. vivax were screened, and 384 were enrolled.”
Case 4 (Line 272-276):
“A 20-year-old man was enrolled in a multicenter clinical trial to assess the noninferiority of a short duration of high-dose primaquine (7 days of 1 mg/kg/d vs. 14 days of 0.5 mg/kg/d) as a radical cure for P. vivax (IMPROV Study, reg no. NCT01814683). A total of 2336 participants from four countries (Afghanistan, Ethiopia, Indonesia, and Vietnam) were enrolled during the period of July 2014-November 2017 [18].”
Case 5 (Line 316-320)
“The subject was a local 13-year-old boy enrolled in a trial assessing the efficacy of spatial repellent to reduce malaria transmission in Southwest Sumba, Indonesia during September 2012-April 2013 [15]. In this trial, 180 G6PD-normal children aged 6-13 years were recruited for presumptive radical cure prior to a 26-week follow-up period as a means of measuring the incidence density of new malaria infections in the community.”
- Case Series 1-5:
- The case series reports mention the years 2012 to 2022. Please mention clearly with references in the introductory lines of each case series whether the findings of these case reports have been published as a part of the trials elsewhere also previously or not.
Response: please refer to no 1
- Several tables have been referred to in the case series but none could be seen in the paper. Please check this and clarify.
Response: all tables are supplementary tables and can be found in the Supplementary Appendices. We have added “Supplementary” to all the supplementary tables to help the readers find them.
- Figure 2 is blurred and not legible. Please replace this with a clear and legible figure.
Response: Thank you for the input. We have improved the quality (page 4-6)
- Abbreviations: It is better to write the complete form followed by abbreviations in parenthesis in the text instead of presenting them under the heading ‘Conflict of interest’. A separate table is not required.
Response: Thank you for the feedback. We have revised it accordingly.
- Discussion:
The G6PD testing methods i.e., SD biosensor (Case 1), CareStart™ RDT (Cases 2, 3, and 4), Binax now RDT (Cases 2 and 3), FST R&D Diagnostics (Case 4) and FST Trinity Biotech qualitative test (Case 5) have undergone a lot of refinement during the recent years after 2012. Please ensure that the discussion considers these facts and limitations and that future suggestions are made accordingly keeping the updated treatment and testing strategies and options in mind.
Response: Thank you for the valuable input. We have added it in the discussion (Line 385-386):
“However, currently available point-of-care G6PD tests represent evolving technologies and improved sensitivity and specificity may yet be achieved [23].”
- References:
Please ensure that all references are as per the journal format.
Response: Thank you for the reminder. We have used the journal format.

Round 2
Reviewer 1 Report
The majority of my comments have been addressed. Just a few minor suggestions:
- Line 65 – suggest replace “solved” with something less definitive, eg. mitigated.
- Line 77 – what does “[AKI]” mean here?
- Line 85 – 86 - I’m not sure it is necessary to state that the authors are colleagues and collaborators, but perhaps it would be more useful to state whether the authors were investigators on the clinical trials?
- Line 115 – this is a trial of PQ for treatment of latent P. vivax, but the authors state that only 12 enrolled subjects had been treated with PQ. Surely more than 12 of the 518 participants enrolled received PQ? Is this a typo?
- Line 129 – suggest delete “slightly” – the word is used multiple times throughout this paragraph and is somewhat subjective and also unnecessary given the absolute values are listed.
- Case 4 – could the authors please include the protocol criteria for treatment cessation, and whether the patient met these criteria (as they have done for Case 1, line 132). Was the patient inadvertently administered PQ, or did this event occur despite adherence to the study protocol?
- Case 5 – line 336 – what was the patient’s Hb and Hillmen score at these time points (days 3 and 4), when he had nausea and epigastric pain? If these were not done, perhaps this could be mentioned (given that the discussion states that all these events occurred in the context of rigorous clinical trial processes and close monitoring).
- Line 407 – I’m not sure that it is necessarily correct to state that in none of the 5 cases were the signs of haemolysis detected early enough to cease PQ any sooner than it was. In case 4, a 5th dose of PQ was given despite a 37% fall in Hb and a Hillmen score of 5. And in case 5, PQ was given in the setting of nausea and epigastric pain without an assessment of Hb or Hillmen score.
Quality of English language is acceptable.
Author Response
Reviewer #1:
- Line 65 – suggest replace “solved” with something less definitive, eg. mitigated.
Response: Thank you for this valuable feedback. We have revised the paper accordingly.
- Line 77 – what does “[AKI]” mean here?
Response: We are sorry but we cannot find [AKI] anywhere in the manuscript.
Line 77: “……problem of access have recently emerged [12,13]. The availability of these in routine care……”
- Line 85 – 86 - I’m not sure it is necessary to state that the authors are colleagues and collaborators, but perhaps it would be more useful to state whether the authors were investigators on the clinical trials?
Response: Thank you for your input. We have amended the text along the lines you suggest (Line 85).
“The authors of this report are the investigators of those trials.”
- Line 115 – this is a trial of PQ for treatment of latent P. vivax, but the authors state that only 12 enrolled subjects had been treated with PQ. Surely more than 12 of the 518 participants enrolled received PQ? Is this a typo?
Response: Thank you for your input. In this trial, subjects were enrolled regardless of their malaria status, and we only gave primaquine to those who were seropositive or had acute malaria, as indicated by the sentence in line 111-112: “Treatment was only given to those with positive serological results and/or having acute malaria.”
Moreover, we have added more information to resolve the confusion (Line 114-115): “Twelve of the enrolled subjects had been treated with primaquine, whereas the remaining 506 were seronegative or in the control arm and were, thus, not treated.”
- Line 129 – suggest delete “slightly” – the word is used multiple times throughout this paragraph and is somewhat subjective and also unnecessary given the absolute values are listed.
Response: Thank you for your comment. We have removed it and also tidied up lines 127 to 129:
“On Day 3, the subject’s Hb level dropped to 10.6 g/dL, a 12% decrease from baseline, his urine colour increased slightly to a Hillmen score of 3 from a baseline of 2, and a third dose was administered.”
- Case 4 – could the authors please include the protocol criteria for treatment cessation, and whether the patient met these criteria (as they have done for Case 1, line 132). Was the patient inadvertently administered PQ, or did this event occur despite adherence to the study protocol?
Response: Thank you for your comment. We have added it (Line 299-300): “The patient was given a fifth PQ dose despite the presence of cessation criteria (fractional Hb fall > 25% from baseline).”
- Case 5 – line 336 – what was the patient’s Hb and Hillmen score at these time points (days 3 and 4), when he had nausea and epigastric pain? If these were not done, perhaps this could be mentioned (given that the discussion states that all these events occurred in the context of rigorous clinical trial processes and close monitoring).
Response: Thank you for the constructive feedback. We have added it (Line 337): “His Hb was not measured and urine color was not checked.”
- Line 407 – I’m not sure that it is necessarily correct to state that in none of the 5 cases were the signs of haemolysis detected early enough to cease PQ any sooner than it was. In case 4, a 5thdose of PQ was given despite a 37% fall in Hb and a Hillmen score of 5. And in case 5, PQ was given in the setting of nausea and epigastric pain without an assessment of Hb or Hillmen score.
Response: Thank you for this feedback. We have revised the text accordingly (Line 408-412) and amended lines 408 and 409 to add clarity.
“The cardinal laboratory sign of acute haemolytic anaemia is a ‘steep’ drop in measured Hb levels in blood that may be accompanied by symptoms of anaemia and/or signs like dark urine, pale conjunctivae and yellow sclera. In three cases, these signs were not detected early enough to prompt a decision to stop PQ, whilst in the other two patients, the clinical course may have been different if PQ had been stopped earlier, as mandated by the protocol (Case 4), or had extra vigilance been exercised by re-assessing the Hb concentration and urine colour (Case 5).”
- Our co-investigator (AWS and PBSA) recently brought to our attention that Subject #5 had indeed undergone genotyping, yielding the result of Vanua Lava. Consequently, we have made the necessary revisions to our paper in light of this recent news:
Line 360-363: Genotyping result showed Vanua Lava variant which turned out to be the dominant G6PD variant in this population [9].
Line 440-442: All of the 4 known variants involved in 5 cases in this series – Coimbra, Union, Viangchan, and Vanua Lava – were of that phenotype.
